# An ISM Approach for Managing Critical Stakeholder Issues Regarding Carbon Capture and Storage (CCS) Deployment in Developing Asian Countries

**Muhammad Ridhuan Tony Lim Abdullah [1,\*], Saedah Siraj [2] and Zulkipli Ghazali [1]**

1   Department of Management and Humanities, Faculty of Science and Information Technology, Universiti Teknologi PETRONAS, Seri Iskandar 32610, Malaysia; zukipli_g@utp.edu.my
2   Faculty of Education, University of Malaya, Kuala Lumpur 50603, Malaysia; saedah@um.edu.my
\*   Correspondence: ridhuan_tony@utp.edu.my

**Abstract:** Carbon capture and storage (CCS) technology deployment in developing Asian countries largely depends on public acceptance, which is highly dependent on the stakeholders involved in CCS. This paper illuminates how stakeholder issues could be strategically managed in the deployment of CCS, in a manner customized to such developing countries. Based on the input from 28 stakeholders of various interests and nationalities (i.e., from China, Malaysia, Thailand, Vietnam, the Philippines, and Indonesia), this study applies Interpretive Structural Modeling (ISM) and MICMAC analysis, in order to develop a management model to address stakeholder issues regarding the deployment of CCS. Our findings revealed eight legislative issues, four social issues, three economic issues, five technological issues, and five environmental management issues. The model revealed that legislative issues, such as those relating to $CO_2$ definition, licensing, land acquisition framework, and expertise, should be managed prior to other issues, that is, in the early stage of CCS deployment. Addressing environmental issues related to promoting public awareness and perception of CCS benefits are among the key drivers in deploying CCS. The study may serve as a reference for CCS deployment in developing Asian countries.

**Keywords:** carbon capture and storage (CCS); interpretive structural modeling; stakeholder management; Asian countries

## 1. Introduction

Development and industrial activities have contributed to increased levels of $CO_2$ emissions globally. Carbon capture and storage (CCS) has been developed and utilized worldwide, in order to minimize $CO_2$ discharges, driven by the concern for global environmental sustainability [1]. As of 2019, there were 19 operational large-scale CCS facilities worldwide, with a total capture capacity of 39.2 million tons per annum (Mtpa), compared to 31.2 Mtpa in 2017 (Global Carbon Capture and Storage Institute, 2019). However, the use of such technology is largely dominated by developed countries, as compared to developing and underdeveloped countries [2]. Due to the rapid economic growth in key regions among developing countries, recent estimates have stated that these countries collectively will contribute to 59% of the world's energy growth and 94% of its coal consumption increase, which would account for an alarming 91% of the global emissions of greenhouse gases (GHG) by the year 2030 [3]. Bream [4] argued that it is crucial to deploy carbon capture and storage (CCS) technology in key developing countries, in order to mitigate this trend. As such, Article 6 of the Paris Agreement through the Green Climate Fund (GCF) allows developing countries to establish local CCS projects at minimal costs, through the use of joint initiatives (p. 37). However, the adoption of CCS in developing countries has faced considerable challenges [3]. Among the most important concerns are the economic growth priority of these countries, compared to their needs to reduce GHG emissions.

Besides economic concerns, public perception, environmental impact, health and safety issues, policy issues, lack of technology, available expertise, and high cost are among the issues commonly faced by developing countries in adopting CCS. Past studies have dealt extensively with these issues in selected aspects, for instance, recent technology used for CCS, the impact of CCS on the environment, health and safety, economic studies, and public acceptance, to name a few. Studies focused on CCS adoption, based on relationships among issues or barriers, are still rare. Hence, contributing to the gap in this aspect, this study aims to develop a framework, from a holistic standpoint, for managing these issues more systematically and strategically, in order to facilitate the sustainable deployment of CCS in developing countries. Rather than taking a reductionist view in exploring the impact of the issues on CCS deployment, a holistic point of view was adopted, considering the relationships among the issues to develop a model based on key stakeholder inputs as a potential guide to deploy CCS technology in developing Asian countries, as the focus of this study.

## 2. Problem Statement

Ashworth et al. [5] have highlighted that, among the critical barriers of CCS deployment, public acceptance is arguably the main determinant for successful CCS projects. Liang and Reiner [6] presented some of the major risks associated with public perceptions against CCS deployment, such as the high cost of $CO_2$ capture, the volatile carbon market, health and safety issues, ecological impacts on the environment leading to adverse uncalculated impacts on societal well-being, and policy-related risks.

Recent public perception studies have revealed significant relationships between public awareness and knowledge of carbon capture, utilization, and storage (CCUS) and their perception. For example, Yang et al. [7], in their study focused on 27 Chinese provinces, revealed that 51% of the respondents were against the technology, due to their lack of basic awareness and knowledge regarding the potential benefits of CCS. The lack of interest in climate change was also a contributing factor to such low acceptance. In another study, Chiang and Pan [8] reported that 94% of the total respondents (N = 679) were highly educated, with an overall acceptance of CCS development of 90%. This was attributed to the awareness and knowledge of CCS of respondents (being university graduates), as well as their awareness of climate change. This finding has been supported by a later study of Li [9], who reported that 74.3% of the tertiary-educated respondents supported CCS, while 91.4% of them agreed with climate change issues. Similar findings have also been reported in similar studies in Europe. For instance, a low public knowledge of $CO_2$ technologies led to biased perceptions of CCS and CCU (carbon capture and utilization) [10,11]. These findings indicated that the two main prerequisites for public acceptance of CCS were public awareness of climate change and knowledge regarding CCS or CCU in addressing $CO_2$ emissions [12].

Recent studies have also pointed out specific factors affecting public acceptance, in terms of health and safety. A phenomenon, known as the NIMBY (not in my backyard) effect, describes that being personally affected by CCS would evoke higher risk perception among the public; this, in certain cases, brought about the rejection or cancellation of $CO_2$-related technologies [13]. For example, Arning et al. [10] conducted a study on public perception and acceptance of CCUS in Germany, and reported that initially high support from the public towards CCUS dropped significantly when respondents became affected by CCS or CCU sites hypothetically being developed in their neighborhood. Similar findings have been supported by studies in China. Chen [12] stated, in their findings, that although 90% of their respondents did not object to CCS in China, over 50% of the respondents were not willing to have $CO_2$ pipelines running through or very near to their locality. Similar results from another study in China showed that, although 80.4% of participants believed that CCUS may help to mitigate the impacts of global warming, the NIMBY phenomenon was obvious from the location-based objections to transportation and storage processes [14].

In favor of public acceptance towards CCS projects, Yang et al. [7] stressed the importance of gaining public trust through involved stakeholders, such as scientists, environmental organizations, government institutions, and energy companies, in their risk and benefit perceptions. As public perception is largely affected by their knowledge on CCS deployment, the public reliance on stakeholder views is especially evident when it comes to emerging technologies, as the public relies on stakeholders who are actively involved in the solutions for trusted information [15]. Studies have also shown that public perception is significantly affected by the way stakeholders influence policy [16,17]. Accordingly, public trust also stems from their belief that stakeholders have the responsibility of aiding in addressing social problems [18]. Thus, it is important to involve stakeholder views prior to the deployment of CCS initiatives, especially those who have gained public trust. However, stakeholder perceptions are shaped by their view on CCS issues, which would then influence public perceptions. Table 1 lists existing studies in the areas relating to CCS issues based on stakeholder perception pertaining to CCS deployment.

**Table 1.** Studies assessing stakeholder perceptions on CCS technology deployment.

| Category | Issues | References |
|---|---|---|
| Economic | High capital cost for CCS installation.<br><br>Lack of financial support from investors and government.<br><br>Increase in gas production cost and selling price due to CCS implementation. | Narita and Klepper [19]<br>Wilberforce, Baroutaji, Soudan, Al-Alami, and Olabi [20]<br>Fridahl and Lehtveer [21]<br>Li, Jiang, Yang, and Liao [22]<br>Narita and Klepper [19]<br>Compernolle, Welkenhuysen, Huisman, Piessens, and Kort [23] |
| Environmental | Risks of $CO_2$ leakage on ecosystem.<br><br>Waste management issue in handling CCS construction waste.<br>Impact on ecosystem and biodiversity due to CCS installation.<br>Groundwater and surface water contamination due to $CO_2$ storage. | Narita and Klepper [19]<br>Yang, Zhang, and McAlinden [7]<br>Larkin [24]<br>Oraee-Mirzamani, Cockerill, and Makuch [25]<br>Lausselet [26]<br><br>Shahbazi and Nasab [27].<br><br><br>Yang, Zhang, and McAlinden [7] |
| Legislative | The need for Environmental Impact Assessment (EIA).<br><br><br>Occupational and HSE standards for processes such as CCS, in order to cope with potential hazards.<br>Identifying the regulatory agencies in the CCS framework (e.g., permits, storage standards)<br>Establishment of leakage liability regime for CCS processes.<br>The need to develop capture, transport, and storage standards.<br>Clear definition of $CO_2$ (currently, $CO_2$ is not defined as a pollutant, contaminant, natural resource, or greenhouse gas).<br>Regulatory framework pertaining to land acquisition, which may be applicable in the event that the land is acquired for pipelines.<br>Consider community opinions in decision-making process related to CCS implementation. | Chiang and Pan [8]<br>Park, Kim, Seo, Hwang, and Ha [28]<br>Bae, Chung, Lee, and Seo [29]<br>Yanagi, Nakamura, and Komatsu [30]<br><br>Bae, Chung, Lee, and Seo [29]<br><br>Fridahl and Lehtveer [21]<br>Yanagi, Nakamura, and Komatsu [30]<br>Yanagi, Nakamura, and Komatsu [30]<br>Dixon, McCoy, and Havercroft [31]<br><br>Arning, Linzenich, Engelmann, and Ziefle [32]<br><br><br>Yanagi, Nakamura, and Komatsu [30]<br><br><br>Zarraby [33]<br><br><br>Li, Jiang, Yang, and Liao [22] |

**Table 1.** *Cont.*

| Category | Issues | References |
|---|---|---|
| Social | Community health and safety issues due to $CO_2$ leakage and seismic activity. | Li, Jiang, Yang, and Liao [22] |
| | Negative impacts to local community due to land acquisition for CCS. | Arning, Linzenich, Engelmann, and Ziefle [32] |
| | Low community awareness of $CO_2$ emissions, climate change, and CCS. | Yang, Zhang, and McAlinden [7] Li, Jiang, Yang, and Liao [22] Xenias, and Whitmarsh [34] |
| | Low level of trust among community towards authorities and implementer. | Yang, Zhang, and McAlinden [7] Fridahl and Lehtveer [21] Xenias, and Whitmarsh [34] Larkin [24] |
| Technology | The safety of $CO_2$ storage for the surroundings can never be sufficiently guaranteed. | Yang, Zhang, and McAlinden [7] Pihkola, Tsupari, Kojo, Kujanpää, Nissilä, Sokka, and Behm [35] |
| | Limited suitable site (onshore and/or offshore) for $CO_2$ storage. | Fridahl and Lehtveer [21] Pihkola, Tsupari, Kojo, Kujanpää, Nissilä, Sokka, and Behm [35] |
| | Immaturity in the development of research and technology base for CCS technology (e.g., technological readiness). | Fridahl and Lehtveer [21] |
| | Lack of CCS expertise. | Bae, Chung, Lee, and Seo [29] |
| | High energy required for CCS processes, which affects the overall energy efficiency. | Wilberforce, Baroutaji, Soudan, Al-Alami, and Olabi [20] |

Based on Table 1, the stakeholder perceptions on CCS technology deployment, by and large, converge to an almost similar focus, related to the consequential outcomes of the technology with respect to economic, environmental, social, legislative, and technological issues. To further elaborate on some of the issues highlighted in Table 1, Wennersten et al. [36] stressed the importance of the economic and social aspects of the technology. As an example, a CCS project in Western Finland was suspended due to insufficient financial support [37]. Consequently, stakeholders started to pull out of the project, which led to its cancellation. Hence, economic and political uncertainties have been identified as the most significant factors in determining the success of a CCS implementation. This point is especially evident in developing countries. For example, in a study on the feasibility of CCS adoption in Vietnam, financial incentives and climate policy were revealed as the most crucial factors for CCS development [38].

In another example from the Barendrecht case study, Shell planned to establish an onshore CCS demonstration project by injecting a huge amount of $CO_2$ into the depleted gas fields located two to three kilometers from the Barendrecht community in 2007. However, due to health and safety issues, the local stakeholders, including the municipal government, the public, and environmental organizations, strongly opposed the project, leading to its cancelation [39]. Another pertinent point to be highlighted is the relationship among the issues discussed here, which amount to significant hindrances in CCS adoption. In the case of the Western Finland CCS project, it was the result of the high CCS installation cost and a lack of governmental support which affected the acceptance of stakeholders in continuing the project. The Barendrecht case of deploying CCS, on the other hand, was the outcome of aggregate issues—a lack of suitable technology use, as well as its potential harmful impact on the environment and the well-being of society—which led to the project being canceled. Relevant to this case, a better choice in technology for CCS which does not harm the environment, with low installation cost, and which is beneficial to society, would have resulted in positive consequences for Barendrecht. For instance, Bioenergy with CCS (BECCS), which uses fermentation to produce bioethanol, would be relatively safer and would be cheaper to develop [40]. In fact, this technology is more relevant in developing countries, which are responsible for approximately 35%

of global ethanol production [41]. Another safer technology would be the application of $CO_2$ conversion technology using direct photoreduction through sunlight, although this technology is currently improving its efficiency issue regarding solar energy exploitation in the photoreduction process. Fortunately, a new breakthrough in such photoreduction technology, by Spadaro et al. [42], has revealed the practical feasibility of $CO_2$ exploitation from exhaust emissions, thus inspiring further studies to find more effective photocatalytic materials (pp. 445–453). These technological findings in CCUS would be more conducive to the acceptance of stakeholders, which would then affect public perception positively towards CCS deployment. In short, for an emerging and largely unfamiliar technology such as CCS to be publicly accepted, stakeholder concerns and issues about the project could significantly become a main deciding factor for deployment. This has been supported by Fedoseev and Tcvetkov [43], who highlighted that CCS communication should be an informed, open, and objective public discussion process, as a vital platform to exchange different public views, especially from the stakeholders on the CCS technology.

Hence, stakeholder views are crucial to the successful deployment of CCS. This includes considering their sensitivity towards the social, economic, and environmental issues of CCS implementation. However, different stakeholders may have different issues, which could be similar, contradict, or differ from one another [44], with respect to CCS deployment. Past studies have extensively investigated the concerns or issues among stakeholders towards the deployment of CCS in their respective countries, as presented in Table 1. However, these studies have largely focused on a particular issue or a set of issues, under a respective domain, although issues are more dynamic, complex, and inter-related with one another. In addition, in the context of developing Asian countries, stakeholders have different views on what aspects are relevant and which aspects have more significant relationships than the others. They would also have different views on how exactly the relationships of these aspects were connected, considering the role of the aspects (either as main drivers or challenges). For example, while investors would view CCS deployment in terms of profits and losses, a non-governmental organization would view it in terms of potential threats to public livelihood. A technologist would be able to advise on the feasibility of deployment while maintaining safety, while an economist would be able to understand whether the benefits of the project outweigh the capital invested. Failure to mitigate their differences would affect public perception and acceptance towards CCS negatively. Unfortunately, despite the importance of aggregating the views of various stakeholders in CCS deployment at a nationwide scale, there is a wide gap in the literature for regional studies in this aspect. Most of the local studies have either focused on appropriate technology for CCS [45], CCS-related economic studies [46,47], or public perceptions [48]. Hence, it would be valuable to develop a holistic implementation model based on all of these aspects, considering the various stakeholder views representing these aspects. Setiawan and Cuppen [49], in their study on stakeholder perspective on CCS in Indonesia, argued that stakeholder acceptance of CCS should be derived from a complex notion which considers their view based on broader factors, instead of their attitude in isolated variables. Considering this approach, this study aimed to develop a stakeholder issues management framework for CCS deployment in developing Asian countries, based on the elicitation of stakeholder views and their consensus. Based on this aim, we highlight the following questions:

1. What are the key issues for carbon capture and storage technology deployment in developing countries, based on stakeholder views?
2. How can carbon capture and storage issues be managed systematically for sustainable deployment in developing Asian countries?

The specific objectives of this study are as follows:

1. To identify and determine the key stakeholder issues for carbon capture and storage (CCS) deployment in developing Asian countries. For this objective, a panel of stakeholders consisting of respondents from diverse backgrounds of expertise and from different countries were involved.

2.  To understand the interdependent and influential relationships among the CCS deployment issues, as well as to provide managerial guidance in achieving sustainable CCS adoption in developing Asian countries.

## 3. Research Method

In this study, the targeted model consisted of a network of issues highlighted by stakeholders, with regards to deployment of CCS in developing Asian countries. These issues were determined by a panel of stakeholders from various backgrounds and nationalities. However, managing stakeholders requires a complex equilibrium of science, industry, society, and policy [50], in order to make informed decisions on CCS implementation policy. Based on the circumstances discussed above, Interpretive Structural Modeling (ISM) and 'Matrice d'Impacts Croises Multiplication Applique a Classment' (MICMAC) analysis were employed, in order to facilitate our investigation into the relationships among the CCS adoption issues. A structural model could then be extracted, based on the relationships for the intended CCS management model [51,52]. ISM was first proposed by Warfield [52], for the analysis of a complex socio-economic system. Through a process of discussion and analysis, ISM can dissolve complex issues, based on qualitative views of experts, by focusing on two ideas at a time. The output of the ISM process is a visual relationship map among ideas and information. This map (model) reveals the underlying concepts of the issues that are important for experts (stakeholders) to discuss, understand, and make sound decisions, with regards to CCS deployment. ISM has been applied in aiding solutions to various complex issues in past studies. For example, Khan et al. [53] applied the ISM and group problem-solving to elaborate the key barriers to technology transfer for green technology. A similar study on technology transfer has also been conducted, in China, using ISM and the Delphi method to investigate the correlation and gradation of influencing factors for international technology transfer [54]. In another discipline, Li and Yang [14] investigated critical waste factors in office building retrofit projects using semi-structured interviews and ISM. A more recent study has attempted to develop an alternative evaluation tool for energy efficiency and optimization, using a method combining ISM and Analytical Hierarchical Processing [55].

The following sections describe the procedure in detail.

### 3.1. Identify and Determine Key CCS Issues

In response to the first research question, we adopted the nominal group technique (NGT) [56], a qualitative method to identify the stakeholder issues related to carbon capture and storage (CCS) management, based on the views of a panel of stakeholders. Studies using ISM suggest adopting expert opinions through various management techniques, such as brainstorming or the nominal group technique, to determine the variables for the issues at hand [51,55]. Nominal groups pose an advantage, as ideas from all experts could be valued and negotiated among the group, while dominance from any party is usually negated [57]. In a previous application, Yang and Lin [51] adopted Interpretive Structural Modeling to construct a conceptual model which consisted a set of 17 drivers for the implementation of green innovation. They incorporated the NGT among 11 experts, in order to identify the key drivers.

In this study, using NGT through purposive sampling of 28 out of 82 experts (stakeholders) from China, Malaysia, Thailand, Vietnam, the Philippines, and Indonesia, who responded to the invitation to participate in this study. In ISM or NGT studies, 'expert' is the typical term that is used to represent a participant. In Bransford and Schwartz [58], when compared to novices, experts have a larger repertoire of knowledge and experience, which enables them to recognize hidden details within complex cases and process them systematically, allowing for better judgement. Moreover, the stakeholders chosen would be able to represent the public who would be affected by the deployment of CCS technologies. In this study, experts refer to the stakeholders, who were committed either as a member of an agency or institution, or who had professional knowledge, concerns, and interest

in the adoption of CCS technology in their respective country. In terms of sample size used, Janes [59] explained that the participation of 8 to 15 experts in the ISM technique is appropriate for the study, considering the complexity of possible exchanges increases exponentially. However, in this type of study, during the selection of participants, priority should be given to the representativeness of perspectives (expertise background) relevant to the focus of the study, in order to allow for quality input, rather than quantity [52,59]. The experts were selected to represent a wide range of expertise, covering social, environmental, economic, industrial, and legal practice aspects. This ensured that the proposed issues from five distinct dimensions were examined by a diverse group of experts. The rationale of an expert's selection is based on their education level, working field, years of working experience, and their knowledge relative to carbon capture and storage (CCS) technology. In this study, the selection of experts (stakeholders) was conducted according to the following criteria:

- The expert must have at least a bachelor's degree (PhD for academics, bachelor or Master for industries).
- The expert must have at least five years of experience in their respective field.
- The expert has knowledge or working experience related to CCS technology.
- The expert has excellent communication skills.
- The expert has commitment to the duration of the study.

A background summary of the participants of the study is shown in Table 2.

**Table 2.** List of expert members (stakeholders) of the study.

| Nationality | Numbers of (Stakeholders) | Backgrounds |
| --- | --- | --- |
| China | 5 | Academic, Environmentalist (NGOs), Energy supplier, Industrial researcher, Policy maker |
| Malaysia | 5 | Academic, Ministry (energy policy), NGOs, National Energy company, Oil and gas company (PETRONAS), Tenaga Nasional Berhad Research |
| Thailand | 5 | Academic, Ministry of Energy, Environmentalist (NGOs), Energy supplier, Industrial researcher, Policy maker |
| Vietnam | 4 | Academic, Environmentalist (NGOs), Energy supplier, Industrial researcher (Vietnam Institute of Energy) |
| The Philippines | 4 | Academic, Environmentalist (NGOs), Energy supplier, Industrial researcher, Policy maker |
| Indonesia | 5 | Academic, Environmentalist (NGOs), Energy supplier, Industrial researcher (LEMIGAS; R&D for CCS) |

The NGT began with independent brainstorming, followed by three rounds of discussion, where participants negotiated ideas among themselves, which involved merging similar issues, omitting irrelevant ones, and classifying the ideas [56]. The discussion was administered over the online social platform Microsoft Teams. The interactions among experts aided in clarifying and justifying the variables (issues), in order to allow for informed decisions [60]. Each variable was presented, familiarized, and clarified, to allow the experts to make an appropriate judgment on whether to include the variable in the final list [61]. In the final stage of NGT, the final list was given to the experts individually, and they voted by indicating a ranking number for each variable. The ranking used had a scale of one (1) to seven (7), where one indicated the least favorable and seven indicated the most favorable item. The aggregated ranking numbers from the experts resulted in the priority aggregated values (PAV) for each issue. Finally, the issues were ranked, based on the total PAV as shown in Table 3. Issues with the highest PAV were those with the highest priority, which were paired in the subsequent step. The outcome of this step satisfied the first research objective, which was to identify and determine the stakeholder issues relating to carbon capture and storage. The subsequent steps satisfied the second objective, that is, developing the carbon capture and storage management model for stakeholder issues.

Based on the finalized list of issues from the NGT sessions, experts (stakeholders) were then asked to judge the relationships among the listed issues. Firstly, the final list of

issues was delivered to the expert panel for further confirmation. Secondly, all 28 experts were asked to answer, "do you think the variable (issue) 'I' directly influences variable j". However, experts may have different judgements on the same pair-wise comparison of two variables. In order to cope with this, Gan et al. [62] suggested applying the principle of "the minority is subordinate to the majority". The contextual relationships among listed variables were determined if six or more experts reached an agreement. After several discussions, contextual relationships between the variables were developed. The results were represented in a self-interaction matrix.

### 3.2. Determine the Structural Self-Interaction Matrix (SSIM)

This step demonstrated the relationships among the CCS issues (identified from the previous step) based on the pair-wise procedure. The pair-wise relationships were based on any set of two variables (i.e., CCS issues). Based on the ISM procedure, four symbols were used to indicate the direction of the relationship between two variables (indexed by i and j) [62]. The description for each symbol is as follows:

- 'V' for the relationship where issue i affects issue j (when i is paired to j),
- 'A' for the relationship where issue j affects issue i (when i is paired to j),
- 'X' for a mutual relationship, where issues i and j affect each other, and
- 'O' for no relationship between the issues (i.e., the variables are unrelated).

Based on the contextual relationships aggregated through stakeholder views, an SSIM was developed, which is provided in Table 4.

### 3.3. Determine the Final Reachability Matrix

Construction of the reachability matrix was carried out, in order to classify the CCS issues into different levels. This is important for developing the model structure and for purposes of interpretation at the end of the study [59]. This was achieved by converting the SSIM (Step 2) into a binary matrix, in order to develop the initial reachability matrix, by replacing V, A, X, and O with 1 and 0, using the substitution rules below:

If entry (i, j) in the SSIM is V, entry (i, j) in the target matrix becomes 1 and entry (j, i) becomes 0.

If entry (i, j) in the SSIM is A, entry (i, j) in the target matrix becomes 0 and entry (j, i) becomes 1.

If entry (i, j) in the SSIM is X, both entries (i, j) and (j, i) in the target matrix become 1.

If entry (i, j) in the SSIM is O, both entries (i, j) and (j, i) in the target matrix become 0.

After obtaining the initial reachability matrix, transitivity was incorporated to develop the final reachability matrix, as shown in Table 5. The property of transitivity rules that, if a CCS issue i influences another issue j, and issue j influences issue k, then issue i essentially influences issue k. Mathematically, if (i, j) = 1 and (j, k) = 1, then (i, k) = 1. This rule was incorporated as shown in Table 5, where '1 *' denotes the relationship based on this rule.

Another pertinent note is that, in the final reachability matrix (Table 5), the summation of each row yields the driving power of each issue, which indicates how much each CCS issue can influence other issues. Furthermore, the summation of each column yields the dependence power of each issue, which indicates how much each issue can be influenced by others. Accordingly, Issue #6, with driving power 21 (higher) and dependence power 1 (lower), was found to be the dominant issue (refer to Table 5).

### 3.4. Determine the Level Partition of the Reachability Matrix

Based on the reachability matrix from step 3.3, the level partitions for each CCS issue were determined. In this process, the reachability sets and the antecedent sets for each CCS issue were generated, as shown in Table 6. The reachability set consisted of the issue itself and other issues which can help to achieve other issues, while the antecedent set consists of the issue itself and other issues that can help in achieving it; for example, referring to Table 6, the reachability set for Issue #1 consists of issues 1, 2, 4, 9, 10, 11, 12, 13, 14, 19, 20, 21, 22, and 24, as these were the issues that it had influence upon. In comparison, Issue #1

was influenced by issues 1, 3, 5, 6, 7, 8, 15, 19, and 21, which constituted its antecedent set. Determination of the level of the reachability matrix element was carried out through the determination of intersection sets. The junction of both sets was obtained for all variables. In the case of Issue #1, the junction of both its reachability set and antecedent set was issues 1, 19, and 21, that is, the issues shared by both its reachability and antecedent sets. Subsequently, the intersection of these sets was acquired for all the CCS issues, and then the levels of each issues were determined. Issues with the same reachability and intersection sets would occupy the top level in the ISM hierarchy. Top-level issues are issues which do not lead to other issues beyond their own level in the hierarchy. During this level partition analysis, once a top-level issue was determined, it was removed from other issues in a first iteration. On the other hand, CCS issues with the largest reachability set and with the smallest antecedent set would occupy the lowest level (Level 1). In this study, as shown in Table 6, Level 1 in the first iteration was occupied by Issue #6. The same process was repeated until partition levels for all issues were obtained. Table 6 shows the 25 CCS issues with their respective reachability sets, antecedent sets, intersection sets, and levels. The level identification process of these issues was completed in 14 iterations.

### 3.5. Development of the Model

From the final reachability matrix and level partitioning, the structural model was generated, which is also known as a digraph. After removing the transitivity links and replacing the node numbers by statements, the ISM model was generated.

### 3.6. Classification of the Variables (CCS Management Issues) through MICMAC Analysis

MICMAC analysis (Méthode de hiérarchisation des éléments d'un système) was used to further analyze the level partitioning of the final reachability matrix obtained in step 3.4. MICMAC was first developed by Arcade et al. [63]. Mandal and Deshmukh [64] claimed that the goal of MICMAC analysis is to identify the influence level of each issue on other issues, by analyzing the driving power and dependence power of each issue. The "driving power" of an issue refers to the aggregate number of issues it affects, while the "dependence power" of an issue is the total numbers of issues which affect it. Next, based on the driving and dependence powers, we created two-dimensional graphs (called driver dependence diagrams), with the horizontal axis representing the extent of the dependence power and vertical axis representing the extent of the driving power. The driving power and dependence of each element can then be used to construct a driver dependence diagram into four clusters, for the classification of variables to four clusters: autonomous cluster, independent cluster, dependent cluster, and linkage cluster. Cluster details are provided below [55]:

1. Autonomous cluster: Poor driving power and weak dependent power. It is somewhat disconnected from the system.
2. Independent cluster: Strong driving power but weak dependent power. Activities that have a very powerful driving force, called "Main activity".
3. Dependent cluster: Low driving power but high dependent power.
4. Linkage cluster: Powerful driving power and strong dependent power. These activities are unstable, in the fact that any action against these activities will affect others, as well as the impact of feedback on themselves.

The summary of the methodology for this study is shown in Figure 1.

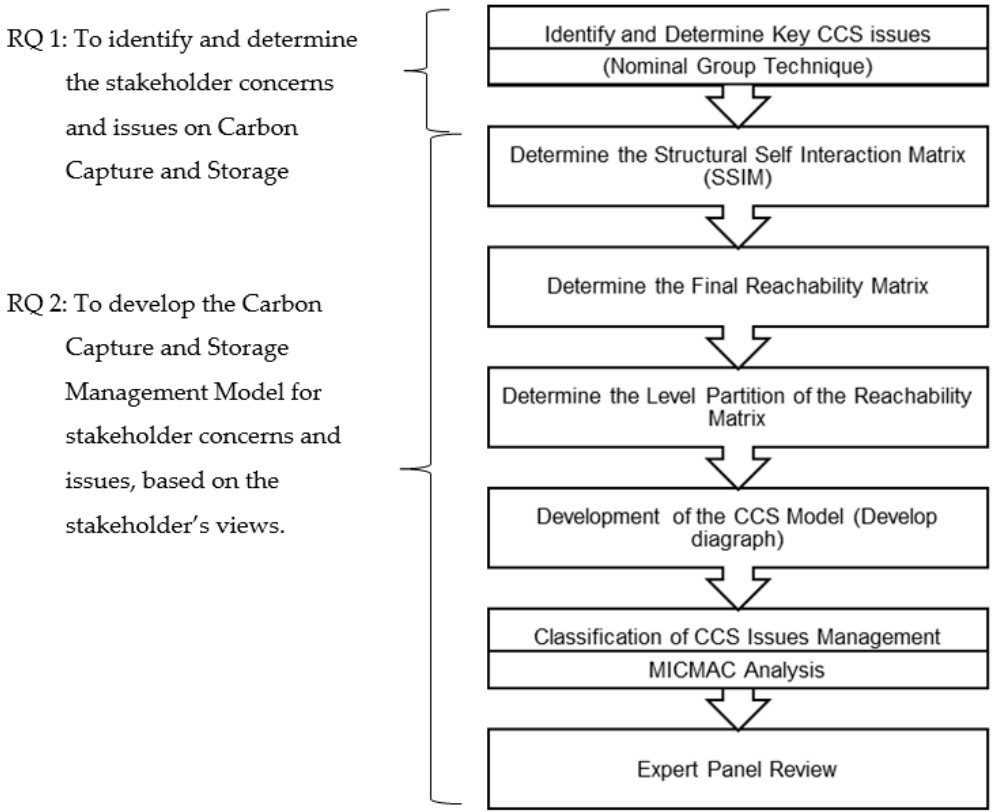

**Figure 1.** ISM methodology flowchart for CCS stakeholder management model.

## 4. Results

In terms of the research questions and the ISM methodology flowchart provided in Figure 1, the results of the study were as follows.

### 4.1. RQ1: What Are the Key Issues for Carbon Capture and Storage Technology Deployment in Developing Countries Based on Stakeholder Views?

Identify and Determine Key CCS Issues

Based on the NGT exercise, Table 3 shows the overall ranking of carbon capture and storage management issues, based on the aggregated priority average value (PAV), which were classified further into five different aspects (i.e., social, environmental, economic, technical, and legislation).

Based on the findings shown in Table 3, the experts identified 24 CCS management issues that relevant stakeholders need to consider in the future deployment of CCS industries and technology in developing Asian countries. The experts (stakeholders) classified the issues into eight legislative issues, four social issues, three economic issues, five technological issues, and four environmental management issues. Based on the ranking (Table 3), legislative issues were identified as the highest priority issues to be managed, compared to other aspects. In this aspect, the need to adapt 'Environmental Impact Assessment (EIA) for CCS implementation' (PAV = 84) led the list, followed by 'Occupational, Health, and Safety standards for processes such as capture, transport, and storage to cope with potential hazards' (PAV = 82) and 'Identifying the regulatory agencies in respective country's framework responsible for issuing permits and ensuring compliance with capture, transport, and storage' (PAV = 80). Following into the ranking were social, technological, environmental, and economic issues. The top aspects for social, environmental, economic, and technological issues were 'Community health and safety issues due to $CO_2$ leakage and seismic activity' (PAV = 68), 'Risks of $CO_2$ leakage on ecosystem' (PAV = 67), 'High capital cost of CCS installation and development' (PAV = 66), and 'The safety of $CO_2$ storage for the surroundings can never be sufficiently guaranteed' (PAV = 63), respectively. The lowest-

ranked issues for these aspects were 'Consider community opinions in decision-making process related to CCS implementation' (Legislative; PAV = 70), 'Low level of trust among community towards authorities and implementer' (Social; PAV = 50), 'Groundwater and surface water contamination due to $CO_2$ storage' (Environmental; PAV = 53), 'Increase in gas production cost and selling price due to CCS implementation' (Economic; PAV = 52), and 'High energy required for CCS processes, which affect the overall energy efficiency' (Technological; PAV = 55).

**Table 3.** Ranking of CCS management issues for CCS management modeling.

| Issues | PAV | R | Aspect |
|---|---|---|---|
| Need to adopt Environmental Impact Assessment (EIA) for CCS | 84 | 1 | L |
| Occupational, Health, and Safety standards for CCS processes to cope with hazards. | 82 | 2 | L |
| Identifying the regulatory agencies in the respective country's framework that would be responsible for issuing CCS permits. | 80 | 3 | L |
| Establish leakage liability regulation for CCS processes. | 77 | 4 | L |
| Need to develop CCS transport and storage standards. | 76 | 5 | L |
| Clear definition of $CO_2$ within respective country's legal framework. | 73 | 6 | L |
| Regulatory framework for land acquisition for CCS implementation. | 72 | 7 | L |
| Consider community opinions in decision-making process for CCS implementation. | 70 | 8 | L |
| Community health and safety issues due to $CO_2$ leakage and seismic activity. | 68 | 9 | S |
| Risks of $CO_2$ leakage into the ecosystem. | 67 | 10 | EV |
| High cost of CCS installation and development. | 66 | 11 | E |
| No sufficient guarantee to surrounding safety due to $CO_2$ storage. | 63 | 12 | T |
| Negative impacts on local community due to CCS land acquisition | 60 | 13 | S |
| Uncertain financial support from investors and government. | 58 | 14 | E |
| Limited suitable sites for $CO_2$ storage (onshore and/or offshore). | 57 | 15 | T |
| Immaturity of research and development in CCS technology. | 56 | 16 | T |
| Lack of CCS experts. | 55 | 17 | T |
| High energy for CCS processes, which affect the overall energy efficiency. | 55 | 18 | T |
| Low community awareness of $CO_2$ emissions, climate change, and CCS. | 54 | 19 | S |
| Waste management issues in handling CCS construction waste. | 53 | 20 | EV |
| Impact on ecosystem and biodiversity due to CCS installation in specific area. | 53 | 21 | EV |
| Groundwater and surface water contamination due to $CO_2$ storage. | 53 | 22 | EV |
| Increase in gas production cost and selling price due to CCS implementation. | 52 | 23 | E |
| Low level of trust among community towards authorities and implementer. | 50 | 24 | S |

Note: L, Legislative; S, Social; E, Economic; EV, Environmental; T, Technological; PAV, Priority aggregated values; R, ranking.

### 4.2. RQ2: How Can Carbon Capture and Storage Issues Be Managed Systematically for Sustainable Deployment in Asian Developing Countries?

4.2.1. Determine the Structural Self-Interaction Matrix (SSIM)

Based on the pair-wise procedure considering the variables (carbon capture and storage issues), a structural self-interaction matrix (SSIM) was developed, as shown in Table 4.

**Table 4.** Structural self-interaction matrix.

| CCSU Issues | 1 | 2 | 3 | 4 | 5 | 6 | 7 | 8 | 9 | 10 | 11 | 12 | 13 | 14 | 15 | 16 | 17 | 18 | 19 | 20 | 21 | 22 | 23 | 24 |
|---|---|---|---|---|---|---|---|---|---|---|---|---|---|---|---|---|---|---|---|---|---|---|---|---|
| 1 | | V | A | V | A | A | A | A | V | V | V | V | V | V | A | O | O | O | X | V | X | V | O | V |
| 2 | | | A | O | A | A | A | A | X | V | X | V | V | V | A | O | A | O | A | V | A | V | O | V |
| 3 | | | | V | V | O | O | V | V | V | V | V | V | V | V | O | O | V | V | V | V | V | O | V |
| 4 | | | | | A | A | A | A | X | V | X | V | V | V | A | A | A | O | A | V | A | V | O | V |
| 5 | | | | | | A | O | X | V | V | V | V | V | V | V | O | O | V | V | V | V | V | O | V |
| 6 | | | | | | | O | V | V | V | V | V | V | V | V | V | V | V | V | V | V | V | O | V |
| 7 | | | | | | | | O | V | V | V | V | V | V | V | O | O | O | V | V | V | V | O | V |
| 8 | | | | | | | | | V | V | V | V | V | V | V | O | O | V | V | V | V | V | O | V |
| 9 | | | | | | | | | | V | X | V | V | V | A | A | A | O | A | V | A | V | O | V |
| 10 | | | | | | | | | | | A | V | V | V | A | A | A | O | A | O | A | V | O | V |
| 11 | | | | | | | | | | | | V | V | V | A | A | A | O | A | V | A | V | O | V |

**Table 4.** *Cont.*

| CCSU Issues | 1 | 2 | 3 | 4 | 5 | 6 | 7 | 8 | 9 | 10 | 11 | 12 | 13 | 14 | 15 | 16 | 17 | 18 | 19 | 20 | 21 | 22 | 23 | 24 |
|---|---|---|---|---|---|---|---|---|---|---|---|---|---|---|---|---|---|---|---|---|---|---|---|---|
| 12 | | | | | | | | | | | | | V | V | A | A | A | O | A | O | A | V | O | V |
| 13 | | | | | | | | | | | | | | V | A | A | A | O | A | A | A | V | O | V |
| 14 | | | | | | | | | | | | | | | A | A | A | A | A | A | A | V | A | A |
| 15 | | | | | | | | | | | | | | | | O | O | O | V | V | V | V | O | V |
| 16 | | | | | | | | | | | | | | | | | A | V | V | V | V | V | O | V |
| 17 | | | | | | | | | | | | | | | | | | V | V | V | V | V | O | V |
| 18 | | | | | | | | | | | | | | | | | | | O | O | O | V | O | V |
| 19 | | | | | | | | | | | | | | | | | | | | V | X | V | O | V |
| 20 | | | | | | | | | | | | | | | | | | | | | A | V | O | V |
| 21 | | | | | | | | | | | | | | | | | | | | | | V | O | V |
| 22 | | | | | | | | | | | | | | | | | | | | | | | A | A |
| 23 | | | | | | | | | | | | | | | | | | | | | | | | V |
| 24 | | | | | | | | | | | | | | | | | | | | | | | | |

## 4.2.2. Determine the Final Reachability Matrix

By replacing the symbols V, A, X, and O with the binary inputs '1' and '0' (according to the rules outlined above), and after checking for transitivity, Table 5 shows the final reachability matrix of the relationships among the issues.

**Table 5.** Final reachability matrix (Conica matrix).

| CC | 1 | 2 | 3 | 4 | 5 | 6 | 7 | 8 | 9 | 10 | 11 | 12 | 13 | 14 | 15 | 16 | 17 | 18 | 19 | 20 | 21 | 22 | 23 | 24 | DP |
|---|---|---|---|---|---|---|---|---|---|---|---|---|---|---|---|---|---|---|---|---|---|---|---|---|---|
| 1 | 1 | 1 | 0 | 1 | 0 | 0 | 0 | 0 | 1 | 1 | 1 | 1 | 1 | 1 | 0 | 0 | 0 | 0 | 1 | 1 | 1 | 1 | 0 | 1 | 14 |
| 2 | 0 | 1 | 0 | 1 | 0 | 0 | 0 | 0 | 1 | 1 | 1 | 1 | 1 | 1 | 0 | 0 | 0 | 0 | 0 | 1 | 0 | 1 | 0 | 1 | 11 |
| 3 | 1 | 1 * | 1 | 1 * | 1 | 0 | 0 | 1 | 1 * | 1 * | 1 * | 1 * | 1 * | 1 * | 1 | 0 | 0 | 1 | 1 * | 1 * | 1 * | 1 * | 0 | 1 * | 19 |
| 4 | 0 | 1 | 0 | 1 * | 0 | 0 | 0 | 0 | 1 * | 1 * | 1 * | 1 * | 1 * | 1 * | 0 | 0 | 0 | 0 | 0 | 1 * | 0 | 1 * | 0 | 1 * | 11 |
| 5 | 1 | 1 * | 0 | 1 * | 1 | 0 | 0 | 1 | 1 * | 1 * | 1 * | 1 * | 1 * | 1 * | 1 | 0 | 0 | 0 | 1 * | 1 * | 1 * | 1 * | 0 | 1 * | 18 |
| 6 | 1 | 1 * | 0 | 1 * | 1 | 1 | 0 | 1 * | 1 * | 1 * | 1 * | 1 * | 1 * | 1 * | 1 * | 1 | 1 | 1 | 1 * | 1 * | 1 * | 1 * | 0 | 1 * | 21 |
| 7 | 1 | 1 * | 0 | 1 * | 0 | 0 | 1 | 0 | 1 * | 1 * | 1 * | 1 * | 1 * | 1 * | 1 | 0 | 0 | 0 | 1 * | 1 * | 1 * | 1 * | 0 | 1 * | 16 |
| 8 | 1 | 1 * | 0 | 1 * | 1 | 0 | 0 | 1 | 1 * | 1 * | 1 * | 1 * | 1 * | 1 * | 1 * | 0 | 0 | 0 | 1 * | 1 * | 1 * | 1 * | 0 | 1 * | 18 |
| 9 | 0 | 1 | 0 | 1 * | 0 | 0 | 0 | 0 | 1 * | 1 * | 1 * | 1 * | 1 * | 1 * | 0 | 0 | 0 | 0 | 0 | 1 * | 0 | 1 * | 0 | 1 * | 11 |
| 10 | 0 | 0 | 0 | 0 | 0 | 0 | 0 | 0 | 0 | 1 | 0 | 1 | 1 | 1 | 0 | 0 | 0 | 0 | 0 | 0 | 0 | 1 | 0 | 1 | 6 |
| 11 | 0 | 1 | 0 | 1 * | 0 | 0 | 0 | 0 | 1 * | 1 * | 1 * | 1 * | 1 * | 1 * | 0 | 0 | 0 | 0 | 0 | 1 * | 0 | 1 * | 0 | 1 * | 11 |
| 12 | 0 | 0 | 0 | 0 | 0 | 0 | 0 | 0 | 0 | 0 | 0 | 1 | 1 | 1 | 0 | 0 | 0 | 0 | 0 | 0 | 0 | 1 | 0 | 1 | 5 |
| 13 | 0 | 0 | 0 | 0 | 0 | 0 | 0 | 0 | 0 | 0 | 0 | 0 | 1 | 1 | 0 | 0 | 0 | 0 | 0 | 0 | 0 | 1 | 0 | 1 | 4 |
| 14 | 0 | 0 | 0 | 0 | 0 | 0 | 0 | 0 | 0 | 0 | 0 | 0 | 0 | 1 | 0 | 0 | 0 | 0 | 0 | 0 | 0 | 1 | 0 | 0 | 2 |
| 15 | 1 | 1 * | 0 | 1 * | 0 | 0 | 0 | 0 | 1 * | 1 * | 1 * | 1 * | 1 * | 1 * | 1 | 0 | 0 | 0 | 1 | 1 * | 1 * | 1 * | 0 | 1 * | 15 |
| 16 | 0 | 0 | 0 | 0 | 0 | 0 | 0 | 0 | 0 | 1 | 0 | 1 * | 1 * | 1 * | 0 | 1 | 0 | 1 | 0 | 1 | 0 | 1 * | 0 | 1 * | 9 |
| 17 | 0 | 1 | 0 | 1 * | 0 | 0 | 0 | 0 | 1 * | 1 * | 1 * | 1 * | 1 * | 1 * | 0 | 1 | 1 | 1 * | 0 | 1 * | 0 | 1 * | 0 | 1 * | 14 |
| 18 | 0 | 0 | 0 | 0 | 0 | 0 | 0 | 0 | 0 | 0 | 0 | 0 | 0 | 1 | 0 | 0 | 0 | 1 | 0 | 0 | 0 | 1 * | 0 | 1 | 4 |
| 19 | 1 | 1 * | 0 | 1 * | 0 | 0 | 0 | 0 | 1 * | 1 * | 1 * | 1 * | 1 * | 1 * | 0 | 0 | 0 | 0 | 1 * | 1 * | 1 * | 1 * | 0 | 1 * | 14 |
| 20 | 0 | 0 | 0 | 0 | 0 | 0 | 0 | 0 | 0 | 0 | 0 | 0 | 1 | 1 * | 0 | 0 | 0 | 0 | 0 | 1 * | 0 | 1 * | 0 | 1 * | 5 |
| 21 | 1 | 1 * | 0 | 1 * | 0 | 0 | 0 | 0 | 1 * | 1 * | 1 * | 1 * | 1 * | 1 * | 0 | 0 | 0 | 0 | 1 * | 1 * | 1 * | 1 * | 0 | 1 * | 14 |
| 22 | 0 | 0 | 0 | 0 | 0 | 0 | 0 | 0 | 0 | 0 | 0 | 0 | 0 | 0 | 0 | 0 | 0 | 0 | 0 | 0 | 0 | 1 | 0 | 0 | 1 |
| 23 | 0 | 0 | 0 | 0 | 0 | 0 | 0 | 0 | 0 | 0 | 0 | 0 | 0 | 1 | 0 | 0 | 0 | 0 | 0 | 0 | 0 | 1 * | 1 | 1 * | 4 |
| 24 | 0 | 0 | 0 | 0 | 0 | 0 | 0 | 0 | 0 | 0 | 0 | 0 | 0 | 1 | 0 | 0 | 0 | 0 | 0 | 0 | 0 | 1 * | 0 | 1 | 3 |
| Total DEP | 9 | 14 | 1 | 14 | 4 | 1 | 1 | 4 | 14 | 16 | 14 | 17 | 19 | 23 | 6 | 3 | 2 | 5 | 9 | 16 | 9 | 24 | 1 | 22 | |

Note: CC, CCSU Management issues; DP, Driving power; DEP, Dependence power; 1 * indicates the values after applying transitivity.

## 4.2.3. Determine the Level Partition of the Reachability Matrix

Based on the findings in Table 5, the level partitioning of the reachability matrix was determined, as shown in Table 6. Based on reachability and antecedent designation for each variable, the partitioning was completed after 14 iterations.

**Table 6.** Level partitioning of reachability matrix.

| CCSU Issues | Reachability Set | Antecedent Set | Intersection Set | Level |
|:---:|:---:|:---:|:---:|:---:|
| | ITERATION 1 | | | |
| 6 | 1,2,4,5,6,8,9,10,11,12,13,14,15,16,17,18,19,20,21,22,24 | 6 | 6 | 1 |
| | ITERATION 2 | | | |
| 3 | 1,2,3,4,5,8,9,10,11,12,13,14,15,18,19,20,21,22,24 | 3 | 3 | 2 |
| | ITERATION 3 | | | |
| 5 | 1,2,4,5,8,9,10,11,12,13,14,15,18,19,20,21,22,24 | 3,5,6,8 | 5,8 | 3 |
| 8 | 1,2,4,5,8,9,10,11,12,13,14,15,18,19,20,21,22,24 | 3,5,6,8 | 5,8 | 3 |
| | ITERATION 4 | | | |
| 7 | 1,2,4,7,9,10,11,12,13,14,15,19,20,21,22,24 | 7 | 7 | 4 |
| | ITERATION 5 | | | |
| 15 | 1,2,4,9,10,11,12,13,14,15,19,20,21,22,24 | 3,5,6,7,8,15 | 15 | 5 |
| | ITERATION 6 | | | |
| 1 | 1,2,4,9,10,11,12,13,14,19,20,21,22,24 | 1,3,5,6,7,8,15,19,21 | 1,19,21 | 6 |
| 17 | 2,4,9,10,11,12,13,14,16,17,18, 20,22,24 | 6,17 | 17 | 6 |
| 19 | 1,2,4,9,10,11,12,13,14,19,20,21,22,24 | 1,3,5,6,7,8,15,19,21 | 1,19,21 | 6 |
| 21 | 1,2,4,9,10,11,12,13,14,19,20,21,22,24 | 1,3,5,6,7,8,15,19,21 | 1,19,21 | 6 |
| | ITERATION 7 | | | |
| 2 | 2,4,9,10,11,12,13,14,20,22,24 | 1,2,3,4,5,6,7,8,9,11,15,17,19,21 | 2,4,9,11, | 7 |
| 4 | 2,4,9,10,11,12,13,14,20,22,24 | 1,2,3,4,5,6,7,8,9,11,15,17,19,21 | 2,4,9,11, | 7 |
| 9 | 2,4,9,10,11,12,13,14,20,22,24 | 1,2,3,4,5,6,7,8,9,11,15,17,19,21 | 2,4,9,11, | 7 |
| 11 | 2,4,9,10,11,12,13,14,20,22,24 | 1,2,3,4,5,6,7,8,9,11,15,17,19,21 | 2,4,9,11 | 7 |
| | ITERATION 8 | | | |
| 16 | 10,12,13,14,16,18,20,22,24 | 6,16,17 | 16 | 8 |
| | ITERATION 9 | | | |
| 10 | 10, 12,13,14,22,24 | 1,2,3,4,5,6,7,8,9,10,11,15,16,17,19,21 | 10 | 9 |
| | ITERATION 10 | | | |
| 12 | 12,13,14,22,24 | 1,2,3,4,5,6,7,8,9,10,11,12, 15,16,17,19,21 | 12 | 10 |
| 20 | 13,14,20,22,24 | 1,2,3,4,5,6,7,8,9,11,15,16,17,19,20,21 | 20 | 11 |
| | ITERATION 11 | | | |
| 13 | 13,14,22,24 | 1,2,3,4,5,6,7,8,9,10,11,12, 13,15,16,17,19,20,21 | 13 | 11 |
| 18 | 14,18,22,24 | 6,16,17,18 | 18 | 11 |
| 23 | 14,22,23,24 | 23 | 23 | 11 |
| | ITERATION 12 | | | |
| 24 | 14,22,24 | 1,2,3,4,5,6,7,8,9,10,11,12, 13,15,16,17,18,19,20,21,23,24 | 24 | 12 |
| | ITERATION 13 | | | |
| 14 | 14,22 | 1,2,3,4,5,6,7,8,9,10,11,12, 13,14,15,16,17,18,19,20,21,23,24 | 14 | 13 |
| | ITERATION 14 | | | |
| 22 | 22 | 1,2,3,4,5,6,7,8,9,10,11,12, 13,14,15,16,17,18,19,20,21,22,23,24 | 22 | 14 |

### 4.2.4. Development of the CCS Model

An ISM model was generated by placing the stakeholder issues (variables), based on their levels, in a digraph, as shown in Figure 2. The issue at level 1 was positioned at the lowest hierarchy in the model; subsequently, the higher-level issues were placed, relatively, at higher hierarchic levels. As mentioned above, issues at the lowest level had the highest driving power, while those at the upper level had relatively low driving power. The model serves as a guide for the management of stakeholder issues related to carbon capture and storage deployment.

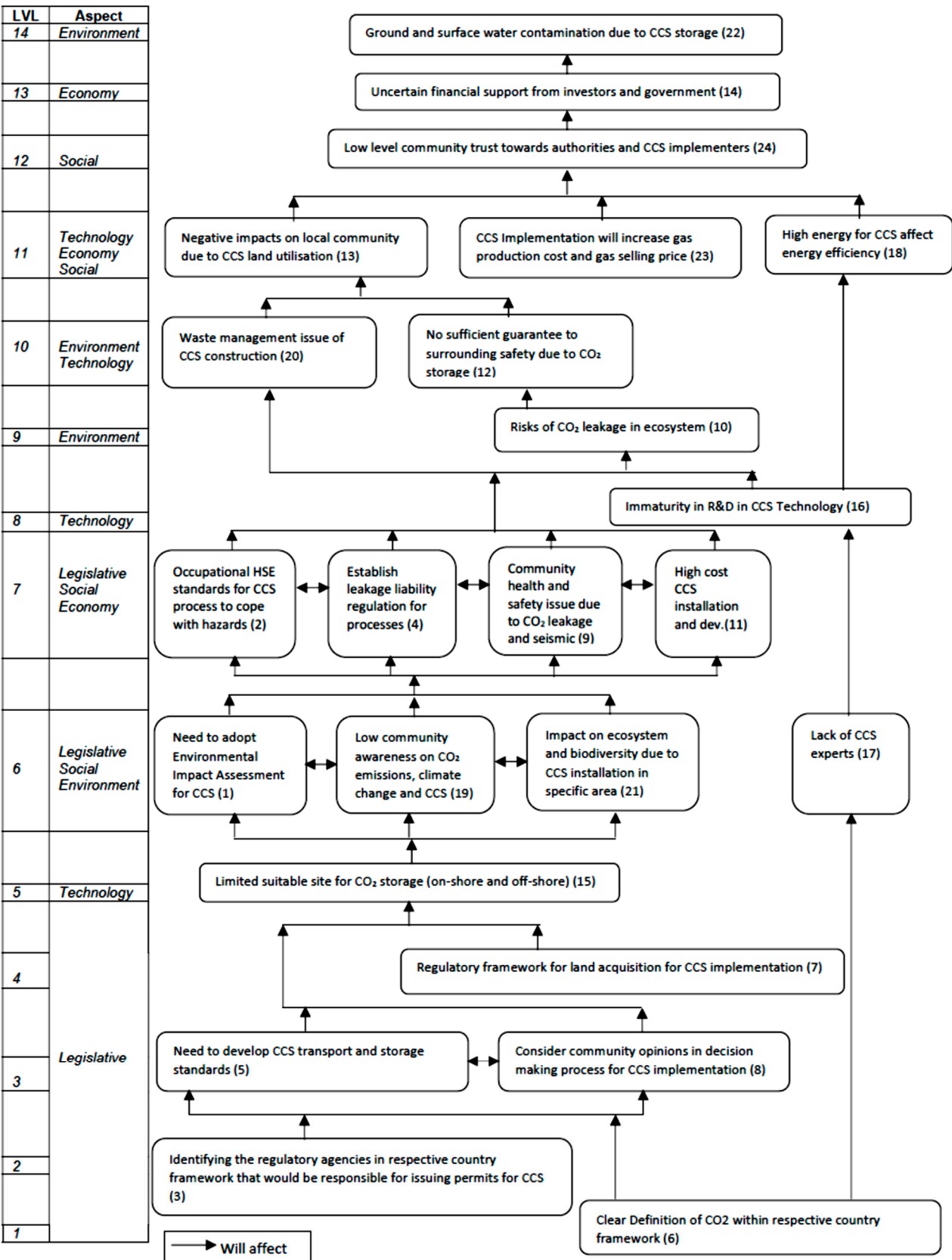

**Figure 2.** Model for managing stakeholder carbon capture and storage issues.

### 4.2.5. Classification of CCS Issues' Management (MICMAC Analysis)

As mentioned in the Methodology Section, MICMAC analysis was conducted to further analyze the degree of influence of the variables (CCS management issues), as shown in Figure 3.

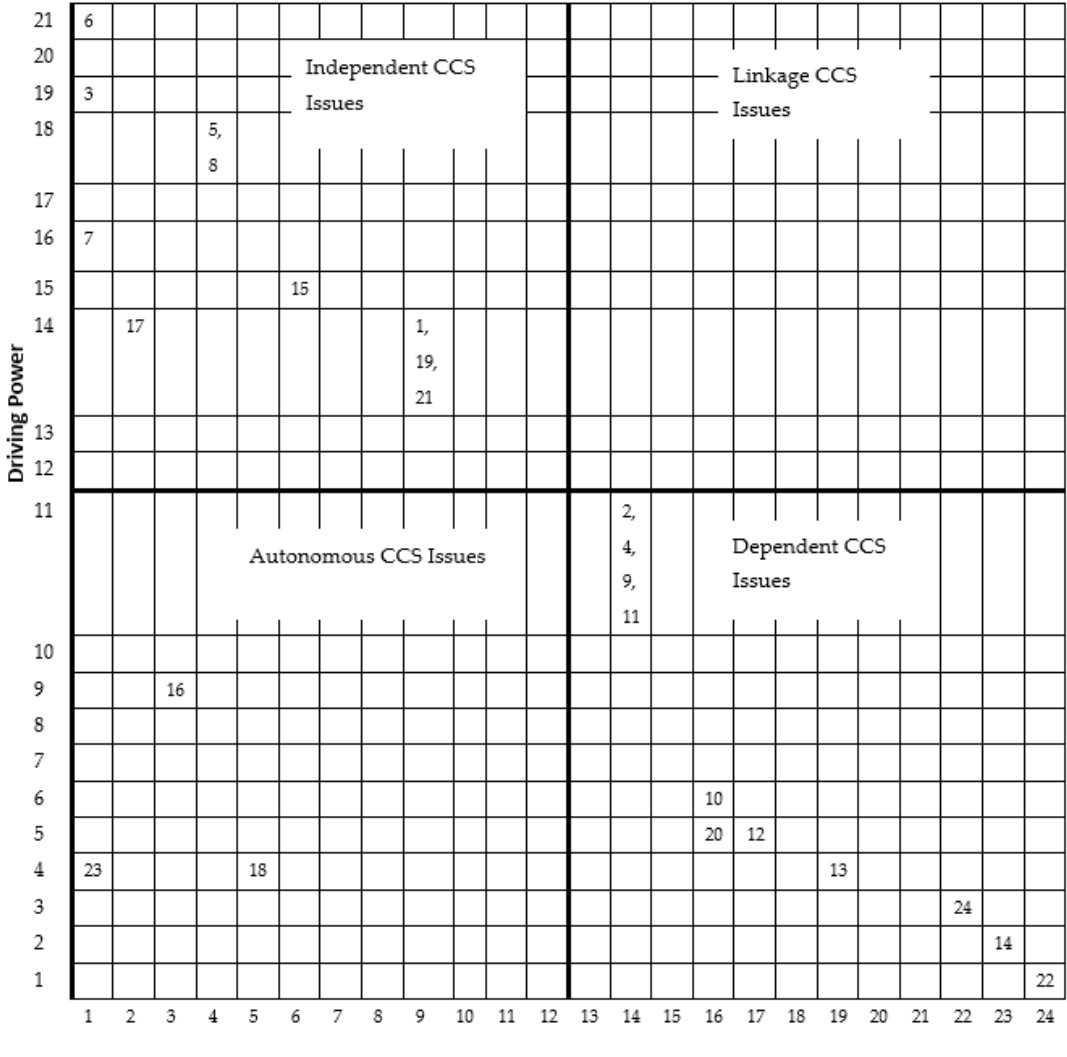

**Figure 3.** Clusters of variables for CCS management model (MICMAC analysis).

Referring to Figure 3, MICMAC analysis revealed how the CCS management issues were categorized, based on their driving and dependence power, as shown in Table 7.

The outcome of the management framework (Figure 2) and the result of the MICMAC analysis (Figure 3) were presented back to the stakeholders (respondents of this study) for their review. Their feedback was compiled and included in the following section for discussion.

**Table 7.** Summary of CCS management issue categories from MICMAC analysis.

| | Cluster | Characteristics | CCS Management Issues |
|---|---|---|---|
| 1 | Independent Variable (Issues) | Important issues which need to be considered prior to other issues | 1,3,5,6,7,8,15,16,17,19,21 |
| 2 | Linkage Variable (Issues) | Issues which serve as a link between independent and dependent variables | None |
| 3 | Autonomous Variable (Issues) | Important issues, but somewhat detached from other issues | 16,18,23 |
| 4 | Dependent Variable (Issues) | Issues which serve to further develop CCS management practices. | 2,4,9,10,11,12,13,14,20,22,24 |

## 5. Discussion

The model in Figure 2 was developed based on the aggregated views of the experts (stakeholders). The first four levels of the model were dominated by legislative issues, which were suggested to be managed prior to other issues for CCS deployment. Through MICMAC analysis, these issues had strong driving power (refer to Figure 3), which consequently led to their being placed at lower levels. The experts identified the issues 'Clear definition of $CO_2$ within respective country's framework' (Issue #6) and 'Identifying the regulatory agencies in respective country's framework that would be responsible for issuing permits' (Issue #3) as main drivers for CCS deployment. During the review session of the model, the expert panel agreed with the outcome and elaborated that a clear understanding of shared definition of $CO_2$ among stakeholders is vital in minimizing future concerns or skepticism of the public on potential nationwide CCS deployment. In the case of 'Identifying the regulatory agencies,' the experts agreed that, within the current regulatory frameworks of developing Asian countries, in terms of CCS adoption, it is of great concern which regulatory agencies should be responsible for issuing permits for CCS projects in developing Asian countries [65]. From the model, managing the issue of developing CCS transport and storage standards (Issue #5) will be manageable under a clear definition of $CO_2$ and with appropriate regulatory agencies in place. Another pertinent point, consistent with the importance of gaining acceptance from the public [7] and considering the NIMBY effect [10], as discussed in an earlier section, the model showed that the CCS transport and storage issue must be managed by taking 'community opinions in decision-making of CCS implementation' (Issue #8) into account, as the model showed that the issues are related. This view is in agreement with a study undertaken by Brunsting et al. [66], who claimed that the absence of the involvement of local stakeholders in the formal decision-making process drove the CCS project in Barendrecht, The Netherlands, to failure. Similarly, Dowd and James [67] emphasized the need to obtain 'social license to operate' (SLO) for CCS projects. The successful management of these four issues, together with the successful regulation of a land acquisition framework (Issue #7), would assist the relevant parties (e.g., government agencies, investors) in overcoming limited site issues for $CO_2$ storage, both onshore and offshore (Issue #15). Besides these issues, CCS deployment also requires suitable CCS experts (Issue #17). The model (Figure 2), again, pointed out that a clear definition of $CO_2$ within the country's context (Issue #6) would help in the selection of suitable experts, based on the country's needs. The experts argued that failure to address these legislative issues would compromise any potential CCS projects. For this reason, the model suggested that this set of legislative issues ought to be managed as priorities, in order to aid in addressing subsequent issues.

These findings were consistent with those of the study undertaken by de Coninck [68], where the authors pointed out that the legal and policy frameworks are vital to the implementation of CCS technology. A trustworthy CCS legal framework is not only acknowledged by relevant stakeholders, it also favors the implementation of the technology [69]. The APEC CCS Regulatory Review [70] has reported that the legal frameworks of developing Asian countries are not ready to address the three variables of CCS: capture, transport, and storage. Hence, this has driven major concerns regarding the establishment

of a comprehensive legal and regulatory framework for future CCS implementation in developing Asian countries [71]. It is crucial to understand the regulatory gaps in the current framework before implementation. This explains why legislative issues attracted the most attention from the experts in this study, as indicated in the model.

Once suitable sites have been determined (Issue #15), further assessment issues need to be in place, in order to validate the sustainable use of sites for CCS storage, as suggested in the model (Issues #1, #19, and #21). The consolidated views of the experts suggested that not only is there a 'need to adapt Environmental Impact Assessment (EIA) for CCS implementation' (Issue #1), but the assessment should also consider addressing the issues of 'low community awareness of $CO_2$ emissions, climate change, and CCS' (Issue #19), as well as the 'impacts on ecosystems and biodiversity due to CCS installation on the site' (Issue #21). An Environmental Impact Assessment is a standard procedure, with the aim of assessing the environmental impacts of a proposed project [72]. As elaborated in the model review session, this ensures that environmental criteria are included in the decision-making process for the undertaken project. In this case, the environmental issues should not be neglected for the sake of CCS implementation. The chain of CCS activities which include $CO_2$ capture, transport, and storage involves the completion of an Environmental Impact Assessment (EIA) before permits can be issued to commence the project [73]. They highlighted that a lack of knowledge on environmental impacts has compromised many CCS initiatives. Therefore, the application of an EIA is more than essential in examining the environmental impacts of a proposed CCS project. The model further indicated that the resolution of the assessment issues (#1, #19, and #21), coupled with managing the issue of a lack of CCS expertise (Issue #17), would contribute to the effective management of 'Occupational HSE standards for CCS process to cope with hazards' (Issue #2), 'Establish leakage liability regulation for processes' (Issue #4), 'Community health and safety issues due to $CO_2$ leakage and seismic' (Issue #9), and 'High cost of CCS installation and development' (Issue #11).

In terms of potential CCS hazards, there are always uncertainties about the potential risks of $CO_2$ caused by leakage. $CO_2$, if present in large amounts, can be toxic. Consequently, the design of occupational, health, and safety standards (Issue #2) is essential, and requires risk management consideration. Furthermore, to cope with the leakage issues, it is important to establish leakage liability regimes for CCS processes. Hence, Issue #4 should be managed together with developing occupational HSE standards, in order to cope with hazards, as suggested in the model. The management of the set of Issues #2, #4, #9, and #11 is essentially important for waste management, especially that resulting from CCS construction (Issue #20). Koornneef et al. [72] highlighted that there would be an increase in waste and other by-products for large-scale power plants which are equipped with CCS technology. The amount of the waste is expected to amount to ten kilotons annually. Thus, it is important to handle this large amount of waste, which is detrimental to the environment (especially without proper management). To further aid in waste management, research and development in CCS technology (Issue #16) could further increase waste management efficiency. Pertaining to Issue #16, the model (Figure 2) revealed a vital aspect where management of immaturity in technology development for CCS is crucial in managing CCS energy efficiency (Issue #18), which then improves community trust towards the authorities and CCS implementers (Issue #24). For example, as highlighted in the previous section, a new pathway in the use of direct photoreduction (solar energy) such as $CO_2$ exploitation from exhaust emissions [42] would be safer and more cost-practical to the environment and society. A safer and more cost-effective technology would increase the likelihood of CCS deployment once the technology is perceived more positively by the stakeholders and the public.

The model also pointed out that "hazard management issues" (Issues #2, #4, #9, and #11), and research and development in CCS technology (Issue #16) would assist in the mitigation of 'Risks of $CO_2$ leakage into ecosystems' (Issue #10), 'the surrounding safety due to $CO_2$ storage' (Issue #12), and 'Negative impacts on local community due to CCS

land utilization' (Issue #13). The management of these issues, together with handling of 'CCS Implementation will increase gas production cost and gas selling price' (Issue #23) and 'High energy required for CCS affects energy efficiency' (Issue #18), are required to mitigate 'Low-level community trust towards authorities and CCS implementers' (Issue #24). Community trust has been identified as a main challenge for the deployment of CCS technology, in almost every stage of its implementation [37]. However, the model revealed that the requirement of "high energy for CCS" (Issue #18) and the "potential increase in cost of gas production and price" (Issue #23), as well as the rest of connected issues, must be resolved prior to winning the trust of communities.

According to Kainiemi et al. [37], post-combustion capture technology is able to achieve a 90% of reduction in the $CO_2$ content of treated flue gases. However, this would lead to an increase in energy consumption, thus reducing the efficiency of the plant by approximately 5%. Junginger et al. [73] also mentioned that more energy would be consumed in the operation of CCS processes, as compared to a plant without a CCS installation. Such energy consumption not only could reduce the overall energy efficiency, it can also have a subsequent impact, increasing the cost of electricity (Issue #23). This extra cost is mainly due to CCS processes such as $CO_2$ capture, compression, transportation, and injection. The increase in the electricity cost could burden society, thus leading to their negative perception. The model supported this claim, where the increase in electricity cost would lead to the community's negative perceptions and trust, which could potentially lead to opposition to CCS projects, to a certain extent.

Community trust towards CCS projects is essential in managing the issues of 'Uncertain financial support from investors and government' (Issue #14) and 'Ground and surface water contamination due to CCS storage' (Issue #22). As CCS projects are capital-intensive, they cannot be handled by the government or private companies alone. Hence, financial support from various sectors is crucial (Issue #14). For this reason, the experts consensually agreed that the issue of "Financial support from investors and government is uncertain" was in the range of important to extremely important, and was placed second on the priority list of economic aspects (Table 3) that need to be resolved at the final stage before CCS deployment. This statement was supported by Kainiemi [37], who reported that governmental financial support was uncertain in a CCS demonstration project launched in western Finland. Due to the very high cost of the demonstration project, without funding from the government, investors pulled out from the project, which slowly led to the cancellation of the project. Thus, regardless of the adoption of any new technology, the issue of uncertain financial support from investors and the government could result in the failure of the project, even at the planning phase.

## 6. Conclusions

Although carbon capture and storage (CCS) technology is an established technology for substantially minimizing the $CO_2$ footprint, it has not been widely adopted in developing countries, compared to developed countries [2]. Public perception has been identified as one of the most critical barriers to CCS adoption and, in the case of developing countries, past studies have substantiated that public perception is significantly influenced by stakeholder concerns and issues towards its deployment. Past studies have largely dealt with public perception or stakeholder concerns and issues relating to CCS deployment. However, these studies were largely confined to specific issues, such as lack of regulatory framework, high CCS costs, impact on public health and safety, and environmental issues, among others, as shown in Table 1. Based on the premise that a certain issue or problem is contributed to by a network of factors or issues [52,59], its solution should be derived from a holistic perspective. Taking this into account, this study used a holistic Interpretive Structural Modeling approach, in order to investigate a framework for managing stakeholder issues systematically and strategically. A total of 28 respondents (stakeholders) from developing Asian countries participated in developing the management framework.

Prior to the development of the stakeholder issue management framework, we identified 8 legislative issues, 4 social issues, 3 economic issues, 5 technological issues, and 5 environmental issues, for a total of 25 critical issues for stakeholders, regarding CCS deployment in developing Asian countries. Through the Interpretive Structural Modeling approach [52], a CCS stakeholder issue management model was developed, which consists of a network of 25 issues at 14 levels (refer to Figure 2). The model substantiates that determining the critical stakeholder issues that hinder the adoption of CCS in a country is not adequate; instead, the relationships among the issues should be mapped out, in order to develop a strategic framework explaining how the issues could be collectively managed to address stakeholder concerns or issues, which then aid in improving public perceptions for successful CCS deployment. The model (Figure 2) revealed that, of the twenty-five issues identified, it is pertinent that, in managing stakeholder CCS issues, legislative issues such as 'Need to develop CCS transport and storage standards' (Issue #5), 'Consider community opinions in decision-making process for CCS implementation' (Issue #8), and 'Regulatory framework for land acquisition for CCS implementation' (Issue #7), driven by the management of 'Clear definition of $CO_2$ within a respective country's framework' (Issue #6) and 'Identifying the regulatory agencies in a respective country's framework that are responsible for issuing permits for CCS' (Issue #3), should be given a primary focus by CCS stakeholders, before other issues could be managed. This was made evident through the MICMAC analysis (Figure 3), in which these legislative issues were classified as 'Independent issues,' due to the higher driving power of their influence in aiding the management of other categories of issues (i.e., technological, environmental, social, and economic issues). For example, the governments of developing Asian countries could enact regulatory frameworks that assure CCS projects are safely and effectively developed and operated. A regulatory framework on the siting of the sequestration of the $CO_2$ onshore or offshore should also be conducted safely, in order to protect communities, the environment, and other resources, especially ground and surface water.

The CCS management model in Figure 2 further showed that management of the legislative issues, as discussed here, would aid in the management of the technological issue 'Limited suitable site for $CO_2$ storage (onshore and/or offshore)' (Issue #15). This issue is subject to the management of legislative issues which are essential in aiding the mixture of other legislative, social, and environmental issues, such as "Need to adopt Environmental Impact Assessment (EIA) for CCS' (Issue #1), 'Low community awareness of $CO_2$ emissions, climate change, and CCS' (Issue #19), and 'Impact on ecosystem and biodiversity due to CCS installation in specific area' (Issue #21). These, then, aid in managing the legislative issues 'Occupational, Health, and Safety standards for CCS processes to cope with hazards' (Issue #2) and 'Establish leakage liability regulation for CCS processes' (Issue #4), the social issue 'Community health and safety issues due to CO2 leakage and seismic activity' (Issue #9), and the economic issue 'High cost of CCS installation and development' (Issue #11). In addition, the MICMAC analysis (Figure 3) revealed that, together with the legislative issues discussed earlier (issues #3, #6, #5, #8, and #7), the set of issues #1, #19, and #21, as well as the set of issues #2, #4, #9, #11, and #17, fell into the category of independent issues, which represented primary driving issues that could hinder the successful deployment of CCS. These issues, when successfully managed, would be essential in managing the 'dependent' issues 'Risks of $CO_2$ leakage into ecosystem' (Issue #10), 'No sufficient guarantee to surrounding safety due to $CO_2$ storage' (Issue #12), 'Waste management issue in handling CCS construction waste' (Issue #20), 'Negative impacts to local community due to CCS land acquisition' (Issue #13), 'Low level of trust among community towards authorities and implementer' (Issue #24), 'Uncertain financial support from investors and government' (Issue #14), and 'Groundwater and surface water contamination due to $CO_2$ storage' (Issue #22), with the aid of the management of 'autonomous' issues: 'Immaturity of research and development in CCS technology' (Issue #16), 'High energy for CCS processes, which affects the overall energy

efficiency' (Issue #18), and 'Increase in gas production cost and selling price due to CCS implementation' (Issue #23).

The findings of this study have implications. In terms of practical implications, we presented a systematic model as a framework to manage stakeholder issues regarding sustainable carbon capture and storage (CCS) deployment, which is applicable to developing Asian countries. The model provides a strategic guide to stakeholders, such as policy makers, energy initiative planners, investors, or environmentalists, in their respective countries, to develop a strategy for managing issues prior to the deployment of CCS. For example, policy makers could focus on the issues with high driving power (independent issues category) for their primary management initiatives towards the deployment of CCS. Subsequent issues in the model could follow suit. In this way, operational costs and time could be effectively and efficiently managed, and optimized in their effort to adopt CCS.

As for academic implications, the study contributes to the theoretical and practical knowledge in the investigation of management of stakeholder issues' analysis. This contributes to the growing concerns among the stakeholders in developing countries, in terms of solutions for the sustainable deployment of CCS, considering that it is a proven technology, not only in reducing carbon footprints but also as a viable renewable source of energy in developed countries. Although past studies have investigated concerns and issues on the adoption of CCS in developing and underdeveloped countries, few existing studies have investigated the solution by focusing on the relationships among the identified issues. With this study, we aimed to close this gap by analyzing contextual relationships among the identified issues, using an Interpretive Structural Modeling approach.

However, the study was conducted under limitations, which could be taken into consideration for future studies. First, the determination of the selected issues was based on priority average values (PAV) determined through the Nominal Group Technique. A more robust technique, such as the Fuzzy Delphi method, could be employed with more time and resources, in order to determine and rank the issues while considering fuzziness factors among the views of stakeholders. The Analytical Hierarchical Process is another option which could be used to select the issues, based on their respective eigenvalues and the consistency values of respondents. As ISM is highly dependent on expert input, the outcome of the model could potentially be improved by conducting path analysis on the relationships among the issues, in order to validate the model. Nevertheless, this study could be replicated, and its findings could serve as an added reference for other developing countries in modeling solutions appropriate to the country's needs, resources, capabilities, and limitations for successful CCS deployment.

**Author Contributions:** Conceptualization, M.R.T.L.A. and Z.G.; methodology, M.R.T.L.A.; software, Z.G.; validation, M.R.T.L.A. and S.S.; formal analysis, M.R.T.L.A. and S.S.; investigation, M.R.T.L.A. and Z.G. resources, M.R.T.L.A. and S.S.; data curation, M.R.T.L.A. and Z.G. writing—original draft preparation, M.R.T.L.A. and Z.G.; writing—review and editing, M.R.T.L.A. and S.S.; visualization, M.R.T.L.A. and S.S.; supervision, S.S.; project administration, M.R.T.L.A. and S.S.; funding acquisition, M.R.T.L.A. and Z.G. All authors have read and agreed to the published version of the manuscript.

**Funding:** This research was funded by the Ministry of Higher Education (MOHE), Malaysia under the Fundamental Research Grant Scheme(FRGS) with the grant number FRGS/1/2018/SS06/UTP/02/1.

**Institutional Review Board Statement:** Not applicable.

**Informed Consent Statement:** Not applicable.

**Data Availability Statement:** Data will be available on request.

**Acknowledgments:** Support from the Ministry of Higher Education (MOHE), Malaysia via the Fundamental Research Grant Scheme(FRGS) is greatly acknowledged.

**Conflicts of Interest:** The authors declare no conflict of interest.

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
