# Peer review of "An ISM Approach for Managing Critical Stakeholder Issues Regarding Carbon Capture and Storage (CCS) Deployment in Developing Asian Countries"

_sustainability, doi:10.3390/su13126618_

Round 1

Reviewer 1 Report

In this paper, the Interpretive Structural Modelling and MICMAC analysis were used to develop a management model for addressing stakeholders’ issues in deployment of CCS. The Authors have presented a qualitative method of identification and analysis of stakeholders' concerns and issues related to CCS projects.

While the research design and presentation of the methods is clear, the Authors didn't formulate questions or hypotheses.  The lack of hypotheses is justified by the qualitative nature of the study, but research questions should be formulated, apart from the objectives of the study.

What is more, the goals and objectives of the study are not quite consistent with problems stated in the introduction (ln. 130-144) - (1) recognition of a diverse group of stakeholders is crucial in CCS initiatives and (2) there is a wide gap in the literature for local studies in this aspect. The study does not address these issues. This should be corrected.

The presentation of the results is clear and discussion - supported by the results and referenced in the literature.

Minor comments:

The online social platform of Microsoft is named Teams (ln. 233)

Names of the clusters in the Fig.3 should be rather: dependent and independent variables.

In my opinion the form of citing used sometimes by the Authors is not fortunate: "the form like MICMAC was developed by [56]. " Names of the cited authors should be mentioned. 
What is more, according to literature, the authors of the method was described in: Arcade J., Godet M., Meunier F., Roubelat F. (1994), Structural Analysis with the MICMAC Method & Actors‘ Strategy with MACTOR Method, AC/UNU Millennium Project Futures Research Methodology, Paris

The Authors use two terms: driving power (e.g. ln 317) and driver's power (ln. 320) - the first one is a proper one.

Table 4 is illegible.

Author Response

Please see the attachment.  We include our response to your comments, manuscript with track changes for your reference to your comments, and the final edited version (after language edit). 

Thank you.

Reviewer 2 Report

The article outlines the need to address the management of various key stakeholders and issues related to various aspects such as social, economic, and environmental issues before implementing a CCS project.

1. The paper has no specific purpose for the article. Authors should provide it in the summary and in the introduction.
2. What are the research questions that led to the formulation of the aim of the article? Authors should present research questions in conjunction with the description of the problem.
3. In the second section "description of the problem", based on the literature review, only a local gap was identified regarding the lack of research in this aspect - this is a bit too little. This section covers only a literature review related to public perception and issues related to concerns expressed by different stakeholders. The problem of "Carbon Capture and Storage Technology Deployment", which was placed in the first place in the title of the thesis, was completely omitted.
4. Section 3.3. Determine the Final Reachability Matrix - "Logic states that for any 3 variables (A, B, C) with a given relationship when: • A has a relationship with B, (written A → B),…." - it is unclear how these provisions have been applied in section 4.2. Findings for Step 3 - Determine the Final Reachability Matrix. What do we mean by the variables A, B, C?
5. Section 3.5. Development of the model - Figure 1 in this figure was to summarize the research scheme. But how does this flow chart relate to the steps in section 4 onwards? Connect is missing.
6. Section 3.5. Development of the model - "Cluster details are described below [48]: 1. Autonomous cluster: Poor driving power and weak dependent power. There is somewhat disconnected from the system. '' - how was the presented division into clusters related to further research, results, and conclusions?
7. Section 4.3. Findings for Step 4- the Level Partition of the Reachability Matrix - Table 6 is completely unreadable. What does "Level (I ... XIV)" mean has not been explained before.
8. Section - ". Based on reachability and antecedent designation for each variable, the partitioning was completed at 14 iterations. " - not clear statement. How did the authors determine the number of iterations? There is no description of the research methodology.
9. The Conclusions section should contain conclusions from the research. The presented conclusions can be formulated without conducting research.
10. The title of the article does not correspond to its content. The authors did not provide any implementation of the capture technology ... but only a description of the concerns of international experts.
11. This study does not add many new
12. Noticed editing errors are marked in the text of the work.

Author Response

(The authors gave the same response as above.)

Reviewer 3 Report

The review of the Manuscript can be found in the document in attached.

Author Response

(The authors gave the same response as above.)

Round 2

Reviewer 2 Report

The authors of the manuscript "Carbon Capture and Storage Technology Deployment Sustainable Model: Managing Stakeholders’ Concerns in Asian Developing Countries" have carefully revised the text in line with the comments in the review. As a result of the introduced amendments, the article gained wider access to the scientific public. The manuscript was thoroughly reorganized, which greatly helped its readability. I recommend accepting "Carbon Capture and Storage Technology Deployment Sustainable Model: Managing Stakeholders’ Concerns in Asian Developing Countries" to print.